# The Relationship between Numbness and Quality of Life

**DOI:** 10.3390/jcm12041324

**Published:** 2023-02-07

**Authors:** Shuhei Nagai, Hidemi Niwa, Yuki Terajima, Hiroki Igari, Young-Chang P. Arai, Toshihiko Yamashita, Toshihiko Taguchi, Masaya Nakamura, Takahiro Ushida

**Affiliations:** 1Multidisciplinary Pain Center, Aichi Medical University Hospital, Nagakute City 480-1195, Japan; 2Department of Orthopedics, Sapporo Medical University Hospital, Sapporo City 060-8543, Japan; 3Yamaguchi Rosai Hospital, Sannyouonoda City 756-0095, Japan; 4Department of Orthopedics, Keio University Hospital, Tokyo 160-0016, Japan

**Keywords:** numbness, painless, quality of life

## Abstract

Background: Numbness is a term commonly used in clinical practice to describe an abnormal sensory experience that is produced by a stimulus or is present even without a stimulus. However, there is still much that remains obscure in this field, and also, few reports have focused on its symptoms. In addition, while pain itself is known to have a significant impact on quality of life (QOL), the relationship between numbness and QOL is often unclear. Therefore, we conducted an epidemiological survey and analyzed the relationship between painless numbness and QOL, using type, location, and age as influencing factors, respectively. Methods: A nationwide epidemiological survey was conducted by mail using a survey panel designed by the Nippon Research Center. Questionnaires were sent to 10,000 randomly selected people aged 18 and over from all over Japan. Out of the 5682 people who responded, the relationship between numbness and QOL was analyzed using the EuroQol 5 Dimension-3L (EQ5D-3L) for patients who are currently experiencing painless numbness. Findings: The results suggest that painless numbness affects QOL and that QOL decreases as its intensity increases. Furthermore, the two factors of numbness of feet and numbness among the young may be less likely to affect QOL. This study may be of great significance in the field of numbness research.

## 1. Introduction

Among the subjective symptoms that plague us along with pain and itching are the sensations known as “paresthesia,” “dysesthesia,” “numbness,” among others. While pain itself has been defined relatively clearly and is widely known by the International Association for the Study of Pain (IASP), the definition of these sensory symptoms remains vague. The terms “paresthesia” and “dysesthesia” are often used with slightly different meanings. In fact, according to the IASP, paresthesia is defined as “an abnormal sensation, whether spontaneous or evoked”. Dysesthesia does not include all abnormal sensations but only those that are unpleasant. However, other glossaries define dysesthesia as a spontaneous abnormal sensation and paresthesia as an induced abnormal sensation or vice versa. The Oxford Dictionary of English defines dysesthesia as “an abnormal unpleasant sensation felt when touched, caused by damage to peripheral nerves”. The word “shibire” in Japanese corresponds to “numbness” in English and includes both abnormal sensations caused with and without tactile stimuli. These are more ambiguously defined than paresthesia and dysesthesia. In clinical practice, many people suffer from these kinds of symptoms in various conditions and diseases, including neurological and spinal cord diseases. Although there have been many discussions regarding the mechanisms and therapeutic medication of this abnormal sensory experience, the lack of a clear definition has hindered research and analysis. Even though there have been studies on the prevalence of numbness and dysesthesia in population studies on diseases such as polyneuropathy, no studies have targeted these symptoms or examined their actual characteristics. Spinal cord diseases, such as lumbar spinal canal stenosis, strangulation neuropathy, and diabetic neuropathy, which are thought to contribute to numbness, increase with age. Previous studies have also shown that nerve conduction velocity decreases with age, suggesting that the normal aging process may also be a factor in the onset of age-related numbness [1]. We thought it was very important to find out how the numbness was affecting the patients. By investigating the impact numbness has on quality of life, both the abnormal sensations that occur in response to stimuli and those that occur without stimuli, the goal is to gain insight into the mechanisms of these symptoms and future treatment. In 2013, the Japanese Orthopedic Association conducted a nationwide epidemiological survey on pain and numbness, including the health and living conditions of the patients. Inoue et al. have already reported the results of the pain-related part of the survey, focusing on neuropathic pain [2]. In this study, among the data not used in previous analyses, we focused on numbness, which is considered useful in the above, and analyzed the relationship between painless numbness and QOL, and the influencing factors for type, location, and age, respectively.

## 2. Materials and Methods

In October 2013, a nationwide epidemiological survey was conducted by mail using a survey panel designed by the Nippon Research Center (Tokyo, Japan). The panel consisted of a random address-based sample with a gender and age distribution similar to that of the Japanese census (http://www.nrc.co.jp/english/services/scheduled.html#panel accessed on 25 December 2022).

Questionnaires (Appendix A) were sent to 10,000 randomly selected people aged 18 and over from all over Japan. The questionnaire used in the study included questions regarding the basic characteristics of the participants (gender, age, occupation, etc.) and questions regarding their pain and numbness (characteristics, severity, location, duration, etc.) (Appendix B).

In particular, the characteristics of numbness were questioned in the following four categories, and the closest answer to one of them was selected.

Unpleasant sensations of electricity, such as “zinging”, “tingling”, or “prickling”, even without doing anything.Abnormal sensations or discomfort when touching.A dull sensation of “thickened” or “leathery” skin when touching.None of the above.

The relationship between numbness and quality of life (QOL) was analyzed using the EuroQol 5 Dimension-3L (EQ5D-3L) for patients who currently feel only numbness without pain. The intensity of numbness was rated on an 11-point scale, with 0 referring to no numbness, 10 referring to intolerable numbness, 1–2 as low numbness, 3–6 as moderate numbness, and 7–10 referring to high numbness. The type of numbness was differentiated between “evoked” abnormal sensations caused by external stimuli and “spontaneous” abnormal sensations caused in the absence of external stimuli. The EQ5D is used worldwide. The EQ5D is based on 243 different health status scores, with “perfect health = 1” and “death = 0”. Higher scores indicate a higher quality of life.

The study was conducted in accordance with the ethical guidelines of Aichi Medical University (Nagakute, Japan).

### Statistics

Data were analyzed using SPSS version 29.0 (IBM, Armonk, NY, USA) for Windows. We analyzed the data as ordinal scales of qualitative variables. The dependent variable was investigated for normal distribution with the Shapiro–Wilk test, with EQ5D as the dependent variable and the factor as the intensity of numbness. Since all groups did not show a normal distribution or the total number was small, the Kruskal–Wallis test (non-parametric test with no correspondence between the multiple groups) was used to examine the significance between the three groups. The Bonferroni test was used to compare the groups for multiple comparisons. The significance level was set at *p* < 0.05 for all analyses. The Bonferroni test used an adjusted significance level.

## 3. Results

Questionnaires were sent to 10,000 people randomly and the results of 5682 returned questionnaires were analyzed. The response rate was 56.8%. Out of these 5682 patients, 768 patients who are currently experiencing numbness without pain were selected.

The relationship between numbness and QOL was analyzed using the EQ5D in 768 patients. The 768 patients were divided into three groups according to the intensity of numbness (low: 214 (27.9%), moderate: 457 (59.5%), high: 95 (12.4%)). Those with blanks were excluded. There was no major difference in patient backgrounds among these three groups (Table 1 and Table 2).

The median values for each group are shown below (low: 0.898 ± 0.782 mod: 0.898 ± 0.811 high: 0.787 ± 0.995). As none of the three groups divided by intensity showed a normal distribution, multiple comparisons were performed according to a non-parametric test with no correspondence between the multiple groups. Kruskal–Wallis found significant differences and Bonferroni found significant differences between all groups. (Shapiro–Wilk: *p* < 0.001 (all groups), Kruskal–Wallis: *p* < 0.001, Bonferroni: *p* < 0.001 (low-mod, mod-high, low-high)). As numbness increased, EQ5D decreased significantly in all groups (Figure 1).

The 768 patients were divided into spontaneous (*n* = 421) and evoked (*n* = 208) by the type of numbness. For each, they were similarly divided into three groups (low, moderate, and high) according to the intensity of numbness (Figure 2).

### 3.1. Spontaneous Group

The total number was 421 (low: 99 (23.5%) mod: 254 (60.3%) high: 68 (16.2%)). The median values for each group are shown below (low: 0.898 ± 0.943 mod: 0.860 ± 0.793 high: 0.794 ± 0.990)

No group showed a normal distribution. (Shapiro–Wilk: *p* < 0.001 (all groups))

The results of the analysis, in the same way, showed significant differences between all groups (Kruskal–Wallis: *p* < 0.001, Bonferroni: *p* < 0.001 (low-mod, mod-high, low-high).

As numbness increased, EQ5D decreased significantly in all groups.

### 3.2. Evoked Group 

The total number was 208 (low: 67 (32.2%) mod: 122 (58.7%) high: 19 (9.1%)). The median values for each group are shown below (low: 0.898 ± 0.582 mod: 0.898 ± 0.861 high: 0.762 ± 0.128).

Low and moderate groups did not show a normal distribution (Shapiro–Wilk: *p* < 0.001). High group shows a normal distribution (Shapiro–Wilk: *p* = 0.177). However, the number is very small at 19. Therefore, we analyzed it in a similar way using non-parametric tests. The Kruskal–Wallis test showed significant differences (*p* < 0.001). Bonferroni test showed no significant differences in moderate-high, but significant differences were found in the remaining two groups (mod-high: *p* = 0.434, low-mod: *p* < 0.001, low-high: *p* = 0.008). As numbness increased, EQ5D decreased in all groups. Not significant, but a trend was observed.

We focused on the extremities, which had the highest prevalence in this study. They were divided into foot (*n* = 162) (Figure A2 in Appendix B-10.11.26.27), hand (*n* = 152) (Figure A2 in Appendix B-16.17.34.35), and other area (*n* = 392). For each, they were similarly divided into three groups (low, moderate, and high) according to the intensity of numbness (Figure 3).

### 3.3. Foot Group

The total number was 162 (low: 61 (37.7%), mod: 85 (52.5%), high:16 (9.9%)). The median values for each group are shown below (low: 0.898 ± 0.780, mod: 0.898 ± 0.849, high: 0.826 ± 0.862).

Low and moderate groups did not show a normal distribution (Shapiro–Wilk: *p* < 0.001). High group showed a normal distribution (Shapiro–Wilk: *p* = 0.305). However, the number is very small at 16. Therefore, we analyzed it in a similar way using non-parametric tests. The Kruskal–Wallis test showed no significant differences (*p* = 0.23).

Bonferroni test showed significant differences in low-high, but no significant differences were found in the remaining two groups (low-high: *p* = 0.032, low-mod: *p* = 0.198, mod-high: *p* = 0.402). As numbness increased, EQ5D decreased in all groups. Not significant, but a trend was observed.

### 3.4. Hand Group

The total number was 152 (low: 47 (30.9%), mod: 92 (60.5%), high: 13 (8.6%)). The median values for each group are shown below (low: 0.898 ± 0.814, mod: 0.898 ± 0.770, high: 0.792 ± 0.594).

Low and moderate groups did not show a normal distribution (Shapiro–Wilk: *p* < 0.001). High group show a normal distribution (Shapiro–Wilk: *p* = 0.365). However, the number is very small at 13. Therefore, we analyzed it in a similar way using non-parametric tests. The results of the analysis, in the same way, showed significant differences between all groups (Kruskal–Wallis: *p* < 0.001, Bonferroni: low-mod *p* = 0.06, mod-high *p* = 0.07, low-high *p* < 0.001).

As numbness increased, EQ5D decreased significantly in all groups.

### 3.5. Other Area Group

The total number was 392 (low: 101 (25.5%), mod: 237 (60.5%), high:54 (13.8%)). The median values for each group are shown below (low: 0.898 ± 0.778, mod: 0.898 ± 0.801, high: 0.775 ± 0.104).

No group showed a normal distribution. (Shapiro–Wilk: *p* < 0.001 (low, mod), *p* = 0.006 (high)). The results of the analysis, in the same way, showed significant differences between all groups. (Kruskal–Wallis: *p* < 0.001, Bonferroni: *p* < 0.001 (low-mod, low-high), *p* = 0.005 (mod-high)) As numbness increased, EQ5D decreased significantly in all groups.

The results of the age-specific trends in EQ5D are shown. The 768 patients were divided according to age: 39 years old or younger (n = 83), 40–69 years old (n = 384), and 70 years old or older (n = 270). In each case, they were similarly divided into three groups (low, moderate, and high) according to the intensity of numbness (Figure 4).

### 3.6. −39 Years Old Group

The total number was 83 (low: 26 (31.3%), mod: 46 (55.4%), high: 11 (13.3%)). The median values for each group are shown below (low: 0.898 ± 0.800, mod: 0.898 ± 0.963, high: 0.898 ± 0.109).

No group showed a normal distribution. (Shapiro–Wilk: *p* < 0.001 (low, mod), *p* = 0.032 (high)). The Kruskal–Wallis test showed no significant differences (*p* = 0.295).

Bonferroni test showed no significant differences in all groups (low-high: *p* = 1.000, low-mod: *p* = 0.538, mod-high: *p* = 0.575). As numbness increased, EQ5D decreased in all groups. Not significant, but a trend was observed.

### 3.7. 40–69 Years Old Group

The total number was 384. (low: 115 (29.9%), mod: 226 (58.9%), high: 43 (11.2%)). The median values for each group are shown below (low: 0.898 ± 0.830, mod: 0.898 ± 0.782, high: 0.820 ± 0.981).

No group showed a normal distribution. (Shapiro–Wilk: *p* < 0.001 (low, mod) *p* = 0.006 (high)) The results of the analysis, in the same way, showed significant differences between all groups (Kruskal–Wallis: *p* < 0.001, Bonferroni: *p* < 0.001 (low-mod, low-high), *p* = 0.006 (mod-high)). As numbness increased, EQ5D decreased significantly in all groups.

### 3.8. 70-Years Old Group

The total number was 270 (low: 71 (26.3%) mod: 166 (61.5%) high: 33 (12.2%)). The median values for each group are shown below (low: 0.898 ± 0.700 mod: 0.837 ± 0.813 high: 0.775 ± 0.966).

No group showed a normal distribution (Shapiro–Wilk: *p* < 0.001 (low, mod) *p* = 0.04 (high)). The results of the analysis, in the same way, showed significant differences between all groups. (Kruskal–Wallis: *p* < 0.001, Bonferroni: *p* < 0.001 (low-mod, low-high), *p* = 0.02 (mod-high)). As numbness increased, EQ5D decreased significantly in all groups.

## 4. Discussion

Numbness is a common symptom in clinical practice. It is generally believed that pain itself is known to have a significant impact on QOL, but numbness itself does not impair daily life [3,4,5].

Concerning numbness related to QOL change, previous studies have reported that numbness in both feet at rest in patients with lumbar spinal stenosis is associated with decreased quality of life and that while numbness has some impact on decreased quality of life, the addition of pain significantly decreases the quality of life [6,7]. However, no large-scale studies have been conducted on numbness-related issues, including QOL changes, to date. One possible reason for this is that the definition of numbness is ambiguous. In this context, we defined numbness as an abnormal, painless, tingling sensation, and classified numbness according to the presence or absence of external stimuli. The present study showed that painless numbness affects QOL and that QOL decreases as its intensity increases.

Our study also found that the prevalence of numbness increased with age (Appendix B Figure A1). Similar to our findings, previous studies have shown that numbness increases in the elderly. According to Inoue et al.’s report [8], the prevalence of numbness was 13.7%, but now that the population is aging, more than 20% of all people experience pain and numbness, and 30% of those aged 75 and older have such symptoms. It is estimated that more than 20% of all people have pain and numbness, especially 30% of those over 75 years of age [9].

Diseases such as lumbar spinal canal stenosis, strangulation neuropathy, and diabetic neuropathy are common causes of numbness, and their prevalence also increases with age.

In the general population, the prevalence of polyneuropathy ranges from 1 to 3%, increasing to 7% in those over 65 years of age [10].

In our study, increased numbness tended to decrease QOL in all age groups. However, detailed results showed that the trend was stronger in the older age group and weaker in the younger age group. It suggests that numbness itself is a factor that decreases QOL. We suggested that differences in prevalence and impact on QOL by age may be related to the causes of numbness. In the elderly, constant organic causes, such as nerve compression associated with bone and ligament deformities and diabetes mellitus, are more common. On the other hand, in the younger, temporary functional cause of numbness, such as postural problems or herniated discs, are more frequent than in the elderly.

Actually, in the present study, 43% of those 70-years old who experienced numbness said “they always had numbness”, while “they always have numbness” was reported by 12% of −39 years old. We expect this suggests a difference in the causes of numbness.

Therefore, we speculate that younger people are not constantly stressed, and therefore, they may be less likely to have decreased QOL.

Gender, an important factor in patient background, was also a consideration but was not analyzed as no major differences were found, as shown in Table 1 and Table 2 (overall: Men (43.9%), women (53.8%), 768 people: Men (44.8%), women (52.1%)).

Concerning numbness location and QOL, in a previous report by Watanabe et al., numbness in both plantar feet in patients with lumbar spinal canal stenosis affected QOL [3]. Weintraub et al. mentioned that foot numbness due to diabetic peripheral neuropathy also affected lower QOL of the patients [11]. As noted above, previous studies on numbness in the legs have been reported and have received much attention. However, our study suggests that numbness in the foot may have less impact on QOL than other sites. We believe that studies focusing on areas other than the foot are needed in the future.

The initial purpose of this study was to investigate the relationship between numbness and QOL. The data used in this study, such as disease type, location, etc., were based on patient reports only, and it is difficult to prove that the information is accurate. Moreover, there were many potential confounding factors in the course of the study.

With these factors in mind, we used simple analysis methods to examine the rough relationship between numbness and QOL. We believe that it is necessary to conduct regression analysis when more reliable data are used in the future.

A limitation of this study is that it is based on patients’ own subjective complaints. In other words, the possibility of duplication or deviation from the actual diagnosis cannot be ruled out. In addition, the question said “Choose one”, which should have been designed differently, to include people with overlap. In the future, it is necessary to collect and analyze accurate data in an environment closer to clinical practice, such as medical examinations. There are still few reports of studies focusing on numbness, and we hope that this study will be useful in clinical practice and help lead to new discoveries.

## 5. Conclusions

Although it is generally believed that numbness itself does not impair daily life, the study shows that painless numbness affects QOL and that QOL decreases as its intensity increases. This also suggested that numbness in the foot and numbness in young people may have little effect on QOL. We believe that studies focusing on areas other than the foot are needed in the future. This study may be of great significance in the field of numbness research.

## Figures and Tables

**Figure 1 jcm-12-01324-f001:**
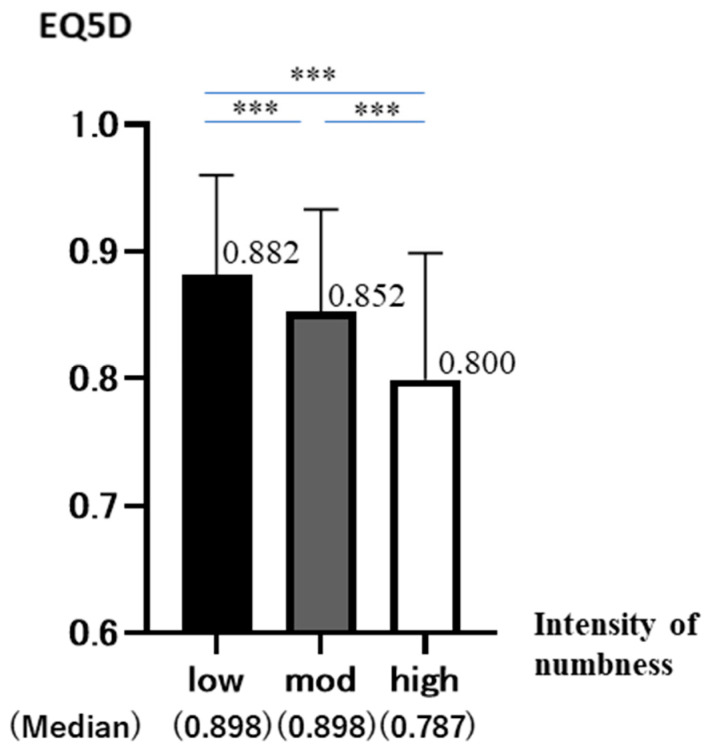
EQ5D is on the vertical axis and numbness intensity is on the horizontal axis (low (*n* = 214), moderate (*n* = 457), high (*n* = 95)). The graph shows the mean and standard deviation (95%CD of EQ5D in each group (*p* < 0.001 ***).

**Figure 2 jcm-12-01324-f002:**
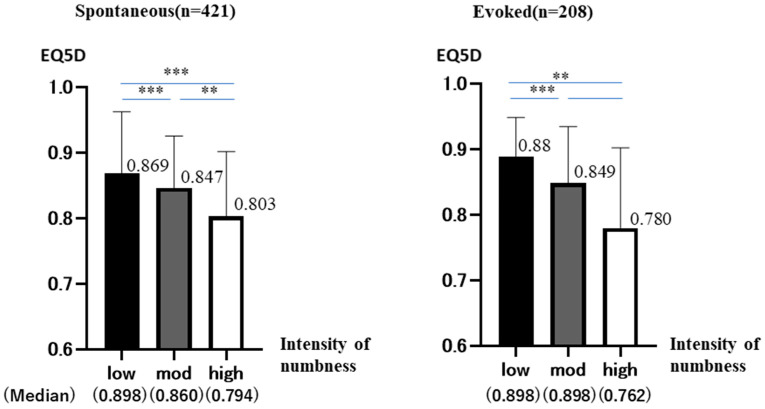
Each type of numbness was analyzed separately as Spontaneous and Evoked. EQ5D is on the vertical axis, and the intensity of numbness is on the horizontal axis. The graph shows the mean and standard deviation (95%CI) of EQ5D in each group (*p* < 0.01 **, <0.001 ***).

**Figure 3 jcm-12-01324-f003:**
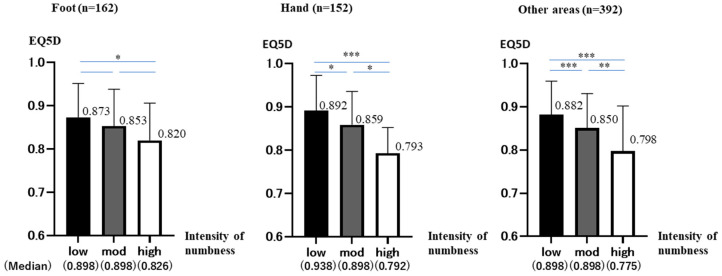
Each site of numbness was analyzed separately as foot, hand, and other areas. EQ5D is on the vertical axis, and the intensity of numbness is on the horizontal axis. The graph shows the mean and standard deviation (95%CI) of EQ5D in each group (*p* < 0.05 *, < 0.01 **, < 0.001 ***).

**Figure 4 jcm-12-01324-f004:**
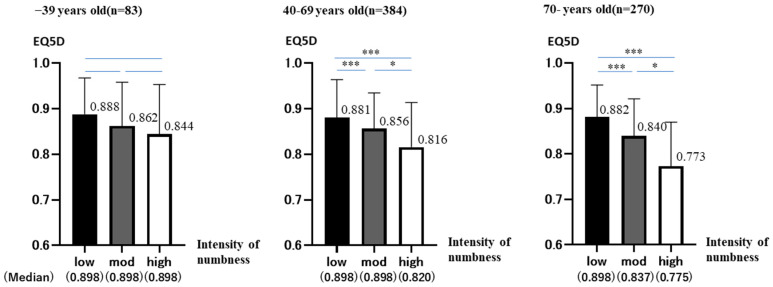
The analysis was divided by age group into 39 years or younger, 39–69 years, and 70 years or older. EQ5D is on the vertical axis, and numbness intensity is on the horizontal axis. The graph shows the mean and standard deviation (95%CI) of EQ5D in each group. (*p* < 0.05 *, < 0.001 ***).

**Table 1 jcm-12-01324-t001:** Background information on the 5682 people who responded to the survey and the 768 people who are currently experiencing numbness without pain.

		Total Number of People (%)	Number of People Currently Numbness without Pain (%)
Total			
		5682	768
Gender			
	male	2493 (43.9)	344 (44.8)
	female	3055 (53.8)	400 (52.1)
	blank space	134 (2.3)	24 (3.1)
Age			
	18–29	559 (9.9)	31 (4.0)
	30–39	822 (14.5)	52 (6.8)
	40–49	941 (16.6)	94 (12.2)
	50–59	790 (13.9)	111 (14.5)
	60–69	1132 (19.9)	182 (23.7)
	70–79	1194 (21.0)	253 (33.0)
	80-	104 (1.8)	22 (2.9)
	blank space	140 (2.5)	23 (3.0)

**Table 2 jcm-12-01324-t002:** Patient background by numbness intensity in 768 people who are currently experiencing numbness without pain.

*n* (%)		Numbness Intensity		
		Low (*n* = 214)	Moderate (*n* = 457)	High (*n* = 95)
Age (mean ± standard deviation)		60.1 ± 14.4	61.0 ± 14.2	59.8 ± 16.7
Gender	Male	92 (42.1)	213 (46.7)	39 (41.1)
	Female	121 (56.5)	229 (50.1)	48 (50.5)
Drinking	Yes	100 (46.7)	223 (48.8)	41 (43.2)
	No	112 (52.3)	231 (50.5)	52 (54.7)
Smoking	Yes	41 (19.2)	91 (19.9)	18 (18.9)
	No	172 (80.4)	362 (79.2)	73 (76.8)
Exercise	Daily	53 (24.8)	122 (26.7)	18 (18.9)
	Once a week	64 (30.0)	127 (27.8)	28 (29.5)
	Once a month	26 (12.1)	41 (9.0)	5 (5.3)
	Once a year	10 (4.7)	14 (3.1)	4 (4.2)
	No	59 (27.6)	143 (31.3)	37 (38.9)

## Data Availability

The data are not publicly available due to that this dataset is the property of the Japanese Orthopedic Association. Please contact us regarding this dataset at the following address: gakusai@aichi-med-u.ac.jp.

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
