# Peer review of "The Relationship between Numbness and Quality of Life"

_jcm, 2023, doi:10.3390/jcm12041324_

Round 1
Reviewer 1 Report
The authors study the possible relationship between abnormal and non-painful somatic perceptions and quality of life. Although the work is simple, I consider that it is sufficiently qualified to be published, taking into account some minor modifications.
- Introduction
The subtle difference between paresthesia and dysesthesia is presented but the fundamental concept of numbness versus paresthesia is not clear in this section.
- Figures 3-6. The horizontal axis should clearly show "numbness intensity".
-Table 6. Missing column headings.
Suggestions
- Table 4. Put it in 2 columns and incorporate it on each side of figure 2.
-Line 118-119. Replace EQ5D with QoL as far as possible.
Reviewer 2 Report
Thank you for this article, which main interest lies in the figures 3, 4, 5, 6. However, all the other parts are quite poor and at least all the major comments below should be adressed. A very significant improvement is thus needed in order to publish this study, which could provide an interesting view about a less commonly adressed topic.
Major comments :
· The presentation is quite strange, with results being mixed in the materials and methods section. Please clarify and re-order (see below).
· Please ensure that all the article is reported according to the STROBE checklist.
· The methods section is quite short, lacks precision, and has major flaws:
ð The study population is not clearly defined, and results are confusing : did the authors focus on people with numbness (with or without pain) or on people with numbness without pain ? According to the objective, all the results should be displayed for people with numbness without pain (e.g., people having had the EQ5D scale). The description of the full set of 1,061 people should be reported only in the appendix section.
ð The statistics paragraph is very short. The methods seem partly inappropriate (except if there was a statistical justification, which was not found). It is even unclear whether quantitative or qualitative variables were analyzed. The threshold for statistical significance is not reported.
ð Was there any « full body numbness » or numbness of multiple body part ? If so, how were they reported ? For these patients, was the numbness intensity the same for all body parts ? It seems important to take into account the « patient » factor. So it could perhaps have been more relevant to classify the numbness as one body part / multiple body parts / full body, for example.
ð The sex factor is not studied. It seems rather strange, since this is a major factor explaining health states.
ð A multinomial logistic regression including all factors studied (age class, numbness site, spontaneous vs evoked numbness, sex) could have been performed to explain a decreased QoL. Please justify why this wasn’t done, at least in the discussion section.
· The reporting of results is quite poor, in both form and content.
Minor comments :
Methods :
· Questionnaires were mailed out and the results of 5,682 returned questionnaires were analyzed. è could you precise the number of questionnaires mailed ? A response rate is expected at the beginning of the results
· The table 1 should be in the « results » section.
· Table 1
o Very basic presentation, to improve significantly before publishing
o Precise the percentages (in column) for each figure
o The figures about numbness type and numbness intensity should be reported, either in both « total » and « people currently numb without pain » columns, or in a separate table dedicated to people currently numb without pain
o Numbness intensity should be reported according to the defined categories : no numbness, low, moderate, etc.
o The figure of people currently numb without pain (n=768) is not coherent with table 2 (n=1,061), probably because table 2 relates to the total number of numb people, with or without pain. Precise clearly what is your study population and analyse only this population. The results about the other one can be reported in the appendix section, if relevant.
· « The EQ5D is based on 243 different health status scores, with “perfect health = 1” and “death = 0”. »
ð precise the version of the EQ5D scale used : 3L or 5L ? it should be 3L according to the 243 health status scores mentioned.
· « Equal variances were confirmed by Bartlett test. »
ð Precise which variables were analysed using this test. It is even unclear whether quantitative or qualitative variables were analyzed.
Results :
· Table 2 and figure 1 give the same information. Please report the table in the appendix section.
· « there was no significant difference in the types of numbness complaints » » è the term « significant » makes reference to a statistical test, which result is not reported here.
· Table 3 : make larger categories in order to make the table more easily readable. The full table could be reported in the appendix section. For example, the authors could separate in three categories : foot / hand / other areas ; as these categories are used in figure 5.
· Table 4 and figure 2 : report each percentage directly on figure 2, in order to make it more easily readable. It could also have been interesting to report the percentages occupied by each body part, among the total body mass/surface. This could show the discrepancy between the percentage of pain felt in one part, compared to the percentage of body mass/surface occupied by this part.
· Table 5 : what are the p-values reported above the table ?
· Table 6 : To clarify this analysis, this sentence in discussion could be reported in the introduction : « Spinal cord diseases such as lumbar spinal canal stenosis, strangulation neuropathy, and diabetic neuropathy, which are thought to contribute to numbness, increase with age. Previous studies have also shown that nerve conduction velocity decreases with age, suggesting that the normal aging process may also be a factor in the onset of age-related numbness [11]. » Moreover, the tests done are not explained. This table is very confusing and difficult to read, whereas figure 6 is clearer.
· « In the foot group, there was no significant difference in EQ5D means between low-moderate groups (p=0.2651) and low-high groups (p=0.0434). » It seems that there is indeed a significant difference in low-high groups.
Discussion :
· The sex factor is not discussed.
· « Although not included in the results column, there were 76 patients with musculoskeletal disease as the 226 causative disease of numbness and pain, and 90 patients with different causes of numbness and pain but not diabetes, stroke, or neurological disease (including other and unknown causes). This meant that 15.6% of those with numbness were considered to have 229 non-neuropathic (musculoskeletal disease, vascular disease, etc.) causes. » è Report these results in the appendix section
Reviewer 3 Report
The authors analyzed the relationship between numbness and quality of life (QOL) in large number of patients using questionnaire. I agree that this study can provide new insight into this area, but I have several major and minor concerns listed as follows:
Major concerns
1) In statistical analysis, the authors had to add test of normality before the Bartlett test in each group.
2) In statistical analysis, the Dunnett test was inappropriate for multiple comparisons: the Dunnett test is adopted to comparison between control group and interventional groups.
3) Did the authors draw the Figure 2 by yourself? If this figure is reprinted material, you have to state it.
4) The Results section contains inappropriate description, which should be described in the Discussion section. Namely, the following descriptions should be deleted from the Results section: “This suggested that low-grade spontaneous numbness may have little effect on QOL (Figure 4)”, “This suggests that numbness in the feet tends to affect QOL but may be less influential than in other areas (Figure 5)”, and so on.
5) Figures and Tables are too many. The authors have to select important materials. I think the Table 2 and 4 should be combined into Figure 1 and 2, respectively. Table 6 should be transferred to supplemental material.
6) The top of Table 5 should present the title of this Table. The description presented in the present version is inappropriate.
Minor concerns
7) Almost all Tables and many of Figures ignore the case. The authors have to check case distinctions.
Reviewer 4 Report
I would like to thank the Editors and Authors for letting me review this manuscript. The authors evaluated data from a previously reported questionnaire based study. They analysed the impact of numbness on quality of life. This is a very important, and often understudied questions, and I commend the authors for looking into this. However, I have some major concerns, particularly relating to the unclear methodology. Most notably, it remains unclear how ‘numbness’ was defined, hence it is not possible to interpret the findings in a clinical context. See more detailed comments below.
The methods are too short and miss important information. E.g., how was numbness recorded? What is the definition/question of numbness, e.g., does it include paraesthesia and dysesthesia (which are gain of function symptoms) or only true loss of function (numbness)? From the discussion it seems that rather than numbness, tingling was included?
I suggest the authors include the questionnaire as an appendix, unless it was a validated questionnaire. Then please provide the name and clinometric properties of the questionnaire.
What was the population studied? What were their diagnoses? How were they selected?
Statistics: please provide more information: what groups were compared?
Results: what if patients had both spontaneous and evoked numbness? This seems not recorded in the cumulative histogram in Figure 1, but surely must have been the case for many?
It is strange that lumbar disc herniation is the most common reason for numbness, as diabetes has a much higher prevalence in the population. So I guess there is selection bias in how the sample was selected. Please make sure you mention how the population was selected.
Table 3: please order by frequency.
What is the difference between cervical radiculopathy and cervical disc herniation?
And between lumbago and lumbar disc herniation?
Where these self-reported diagnoses or tick boxes or clinician reported?
Table 4: again, please sort by frequency
Please explain more how the data derived: e.g., if a patient reported bilateral hand numbness, was this counted twice in frequency or only once?
Analysis of EQ5D with numbness. It is not good statistical practice to subdivide continuous data into groups, as you lose information. Please additionally use the data in a continuous way, e.g., you could do regression/correlation analyses.
Table 5, please also include % for e.g., exercise etc so that it is comparable among groups. This is crucial as the 4 groups were very different in size.
I think one missed comparison is those patients with painful and painless numbness. It would be an important comparison to see whether they are similarly affected or not, e.g, how numbness compares to pain. Can the authors please add that?
Statistical comparison evoked vs spontaneous numbness. Again, how did you deal with those people who had both?
Interpretation: “In the evoked type group, there were signifi-133 cant differences in EQ5D means between all groups (p=0.004, p<0.0001, p=0.0026). This 134 suggested that low-grade spontaneous numbness may have little effect on QOL (Figure 135 4). “ Or would another interpretation be that moderate numbness has no effect on QOL (and the low grade numbness QOL is what is normal)?
Figure 5: please revise the statistical comparisons: it seems that for the Hand, there was no significant difference between moderate and high, even though mean difference was larger and SD tighter than in the ‘others’ area. This may simply be an issue of power (small sample size…)..
The authors run a lot of separate analyses, subdividing groups by age, numbness location, numbness intensity. Why not analysing these data in a model? Also, if so many comparisons are made, did you consider correction for multiple testing?
There is no indication that this study received ethical clearance nor that the participants provided written informed consent to participate. If that is true, then it should not be published.
Round 2
Reviewer 2 Report
Thank you for adressing the comments made. The article has been significantly improved.
Some revisions are still needed (see the revised article).
We also provide some advice to shorten the text of the results in order to facilitate the reading.

Reviewer 3 Report
The pointed concerns were appropriately revised.